# Hypertensive crisis: Insights into prevalence and associated factors at a tertiary care facility in Zambia

Lukundo Siame[1], Mbulazi B. Kabeta[1], Gift C. Chama[1], Lweendo Muchaili[1], Ceaser Wankumbu Silumbwe[1,2], Benson M. Hamooya[1], Bislom C. Mweene[1], Situmbeko Liweleya[1], Sydney Mulamfu[1], Joreen P. Povia[1], Sepiso K. Masenga[1]*

**1** Department of Cardiovascular Science and Metabolic Diseases, Livingstone Center for Prevention and Translational Science, Livingstone, Zambia, **2** Department of Pharmacy, Lusaka Apex Medical University, Lusaka, Zambia

* sepisomasenga@gmail.com

## Abstract

### Background

Hypertensive crisis, including hypertensive emergency (with target organ damage) and hypertensive urgency (without target organ damage), is a critical public health condition at Livingstone University Teaching Hospital (LUTH). Hypertensive crisis has been linked to severe complications, including stroke, renal failure, and heart disease, leading to increased mortality, morbidity, and healthcare costs due to intensive treatment, prolonged hospital stays, and long-term care. This study aimed to determine the prevalence and factors associated with hypertensive crisis among patients presenting at the adult medical emergency department at LUTH.

### Method

This was a retrospective cross-sectional study conducted among 977 individuals aged ≥ 18 years who visited the facility between 1st January and 31st December 2021. Hypertensive crisis was defined as systolic BP ≥ 180 mmHg and diastolic BP ≥ 120 mmHg, with or without target organ damage by the attending clinician. Multivariable logistic regression was used to evaluate factors associated with hypertensive crisis. Statistical significance was set at $p < 0.05$.

### Results

The prevalence of hypertensive crisis was 18.9% (95% CI: 17%, 21%) [(n = 185/977)], with 1.1% (n = 11) diagnosed with hypertensive emergency and 17.8% (n = 174) with hypertensive urgency. The most affected group was under 45 years old (n = 89, 48.1%), and Males [50.8%, (n = 94)] and females [49.2%, (n = 92)] were equally affected. Individuals who did not adhere to their hypertension medication were

**Data availability statement:** All relevant data are within the paper and its Supporting Information files.

**Funding:** This work was supported by the Fogarty International Center and National Institute of Diabetes and Digestive and Kidney Diseases of the National Institutes of Health grants R21TW012635 (SKM), the International Federation of Clinical Chemistry and Laboratory Medicine's Task Force on Outcome Studies in Laboratory Medicine (TF-OSLM) (SKM) and the American Heart Association Award Number 24IVPHA1297559 https://doi.org/10.58275/AHA.24IVPHA1297559.pc.gr.193866 (SKM). The content is solely the responsibility of the authors and does not represent the official views of the National Institutes of Health, IFCC and the American Heart Association. The funders had no role in the study design, data collection and analysis, decision to publish, or preparation of the manuscript.

**Competing interests:** The authors have declared that no competing interests exist.

6.3 times more likely to experience a hypertensive crisis compared to those who adhered (AOR: 6.3; 95% CI: 2.78–13.01; p < 0.001). Individuals in employment were 3.94 times more likely to experience a hypertensive crisis compared to those who were unemployed (AOR: 3.94; 95% CI: 1.52–10.21; p = 0.005). Similarly, individuals diagnosed with a hypertensive crisis had 3.43 times higher odds of being hospitalized than those who were not diagnosed (AOR: 3.43; 95% CI: 1.61–7.34; p < 0.001).

## Conclusion

Hypertensive crisis represents a significant burden on our emergency department, which may lead to severe complications such as stroke, renal failure, and cardio-vascular events. These complications, in turn, result in increased healthcare costs and patient morbidity in resource-limited settings like ours. Therefore, there is a need to enhance public awareness about hypertension and adopt a patient-centered approach to medication adherence.

## Introduction

Hypertension (HTN) is a major public health problem, affecting approximately 1.3 billion adults globally, with the majority living and dying in low- and middle-income countries [1]. Key contributors to the rise include urbanization, physical inactivity, higher levels of stress, higher consumption of salt, sugar, and alcohol, smoking, and limited access to healthcare in low- and middle-income countries [2]. Due to low HTN awareness and control rates, especially in the African region with only 27% receiving treatment and only 12% effectively controlled, the risk of hypertensive crises is increased [1,3]. An estimated 1–2% of people worldwide experience hypertensive crisis, which can present in two forms: hypertensive emergency with specific organ damage and hypertensive urgency with non-specific damage [4]. Hypertensive crisis is emerging as a growing public health concern in sub-Saharan Africa (SSA) [2].

In SSA, the management of hypertensive crisis remains a challenge due to the double burden of infectious diseases and non-communicable diseases (NCDs), which strains already limited healthcare resources [5,6]. Hypertensive crisis presents a serious threat to patient morbidity and mortality [7], resulting in severe complications such as stroke, myocardial infarction, acute kidney injury, and heart failure, requiring immediate medical intervention to prevent disabilities or mortality [8]. While less immediately life-threatening, hypertensive urgencies still require prompt management to prevent progression to an emergency state, which can lead to disability and mortality [9].

In Zambia, as in many other countries in the region, HTN is common with approximately 19% of adults affected and the number is likely to grow while the hospitals are faced with challenges in the management of hypertensive crisis [10,11]. Therefore, this study aimed to determine the prevalence, socio-demographic, and clinical factors associated with hypertensive crisis among patients presenting at the adult emergency department at Livingstone University Teaching Hospital (LUTH).

## Methods

### Study design and setting

This was a retrospective cross-sectional study conducted at the LUTH adult medical emergency department. The department encompasses an outpatient department, an emergency unit, a High Dependency Unit (HDU), and two admission wards (one female and one male). The department serves as a referral center for southern and western provinces of Zambia, the emergency unit receives about 3700–5400 patients annually.

### Eligibility and sampling method

Hospital records were abstracted from participants aged 18 years and above who visited the emergency unit from 1st January to 31st December 2021. Records missing crucial information needed for analysis were excluded (age, blood pressure reading and final diagnosis). The records were chosen using a systematic sampling method where every fifth file from the patient's records in the department was selected for screening and eligibility. The chosen records were subsequently inputted into the research electronic data capture (REDCap).

### Sample size

The estimated minimal sample size was 1007. We estimated a prevalence of 6.3% (p) from a study conducted in Tanzania [4]. We used openepi.com software to calculate the sample size using the formular below, where the alpha level of 1.5%(α),design effect of 1 and population size (N): 1,000,000.

$$n = \frac{DEFF \times N \times p\,(1-p)}{\left(\frac{d^2}{Z^2_{1-\alpha/2}} \times (N-1) + p\,(1-p)\right)}$$

### Variables

The dependent variable in this study was hypertensive crisis, defined as systolic blood pressure of ≥ 180 mmHg and diastolic blood pressure of ≥ 120 mmHg with or without target organ damage [12]. Target organ damage was evaluated by the attending clinician using a combination of clinical history, physical examination findings, and diagnostic tests as per International Society of Hypertension Global Hypertension Practice Guidelines [13]. Evidence of target organ damage included Neurological conditions such as stroke, hypertensive encephalopathy (manifesting as headache, altered mental status, seizures, or focal neurological deficits) [14]. Cardiovascular indicators included acute pulmonary edema (seen as dyspnea, crackles on lung auscultation) as well as myocardial infarction (marked by chest pain, electrocardiogram (ECG) changes, elevated cardiac enzymes and major arrhythmias) [14]. Renal involvement included acute kidney injury (marked by elevated serum creatinine, decreased urine output, proteinuria) [14]. Fundoscopic examination included evaluating for papilledema, cotton wool spots and retinal hemorrhages [14]. The independent variables in the study included age, sex, employment status, chronic kidney disease, diabetes, HIV, history of stroke, hypertensive medication, compliance to medication, marital status, cigarette smoking, alcohol consumption, hospitalization, hypertension status, type of hypertensive crisis and blood pressure.

### Data collection

The data collection exercise was conducted between July 15th, 2022, and February 8th, 2023. Data were abstracted from medical records from patients' latest emergency department visits. Trained research assistants abstracted the data from medical records, and senior data abstractors then reviewed it for completeness and accuracy.

## Data analysis

The data were entered into REDCap and exported to Microsoft Excel 2013. Data was analyzed using STATA version 15. Categorical variables were summarized using frequencies and percentages. Median with interquartile range (IQR) was calculated to summarize continuous variables. To test for normality, the Shapiro-Wilk test was used. Chi-square test was used to determine the relationship between two categorical variables. To determine statistical differences between two medians, the Wilcoxon rank-sum test was used. Multivariable logistic regression was used to evaluate factors associated with hypertensive crisis. Sensitivity analysis was done to assess the potential impact of 79 excluded files on the study findings. The final model's variables were chosen based on published literature and variables that were statistically significant at bivariable analysis. The Hosmer-Lemeshow goodness-of-fit test was used to test for model fitness. Statistical significance was set at $p < 0.05$.

## Ethics

This study was approved by Mulungushi University School of Medicine and Health Science Research Ethics Committee on 19th May 2022 (ethics reference number SMHS-MU2-2022-22). All data collected and analyzed were de-identified to ensure complete confidentiality. No data leading to the identification of the participants was abstracted during the data collection and analysis period. Secondary data were used in this study thus, written/verbal consent was not obtained and was therefore waived by the ethics committee. To strengthen the reporting for this observational study, we used the Strengthening the Reporting of Observational Studies in Epidemiology (STROBE) for reporting (S1 Checklist).

# Results

Out of 5280 files, 1056 were selected systematically and after excluding 79 files due to missing data, a total of 977 were available for analysis, Fig 1.

## Basic characteristics of study participant

The median age in the study was 36 years and participants with hypertensive crisis were order than those without hypertensive crisis (45 vs. 33 year, respectively). 68.8% (n = 672) of the participants were less than 45 years old. Males (n = 463, 47.4%) and females (n = 514, 52.6%) were evenly distributed. The majority were married (n = 554, 57.2%). 55.2% (n = 534) were employed. 3.6% (n = 35) and 2.0% (n = 20) had chronic kidney disease and diabetes, respectively. Additionally, 20% (n = 176) were living with HIV. Only 1.4% (n = 14) of participants had a history of stroke. 236 (n = 24.2%) of the participants were living with hypertension. Among them, 74.2% (n = 178) were on antihypertensive medication, however, the majority (n = 143, 62.7%) were not compliant with medication. Only 8.6% (n = 84) and 31.2% (n = 301) of participants smoked and consumed alcohol, respectively. Nearly half (45.6%, n = 444) of the participants were hospitalized. The median systolic blood pressure and diastolic blood pressure were 126 mmHg (interquartile range (IQR): 112, 143) and 82 mmHg (IQR: 73, 95), respectively, Table 1.

## Relationship between hypertensive crisis and other covariates

The prevalence of hypertensive crisis was 18.9% (95% CI: 17%, 21%) [(n = 185/977)] Table 1. Those who were less than 45 years old were more likely to experience hypertensive crisis than the older age groups (p < 0.001). Individuals with a history of stroke were more likely to experience hypertensive crisis (p < 0.001). Those who were married were more likely to experience hypertensive crisis compared to those who were not (p < 0.001). Among those with hypertensive crisis, participants who were employed were majority compared to the unemployed (p < 0.001). Patients with good antihypertensive compliance were less likely to experience hypertensive crisis compared to those with poor compliance (p < 0.001). Individuals with a history of hypertension were more likely to experience hypertensive crisis compared to those without

**PLOS** **Global Public Health**

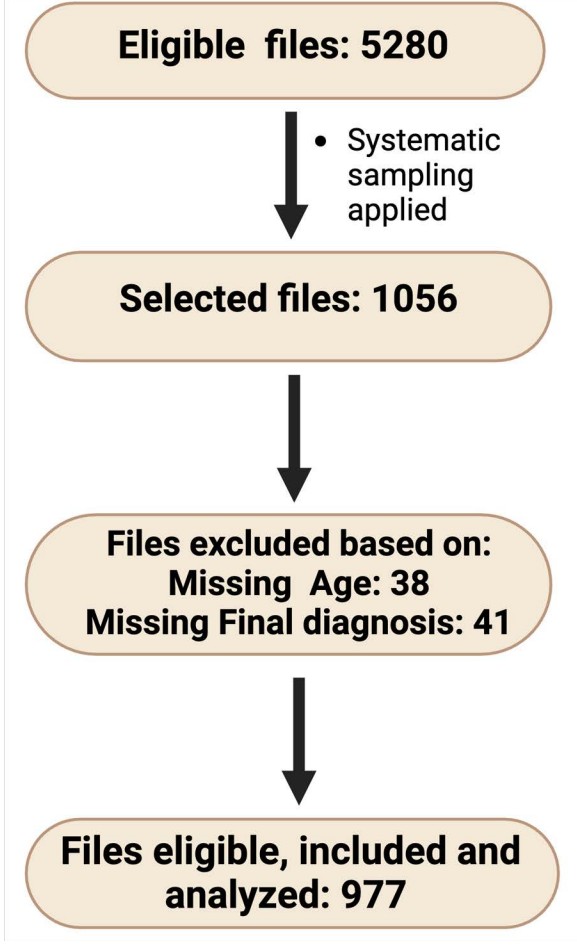

**Fig 1. Flowchart of eligible files.**

hypertension a history (p < 0.001). Participants who were diagnosed with hypertensive crisis were more likely to be hospitalized than those who were not diagnosed (p < 0.001).

**Logistic regression analysis of the factors associated with hypertensive crisis**

At univariable analysis, participants in the age group 45–65 years and above 65 years were 2.70 and 3.67 times more likely to be diagnosed with hypertensive crisis when compared to those aged <45 years, Table 2. Participants with a history of hypertension were 186 times more likely to be diagnosed with hypertensive crisis compared to those without a history of hypertension. Individuals with a history of stroke had 16.7 increased odds of being diagnosed with hypertensive crisis compared to those without a history of stroke. Participants who had poor compliance to antihypertensive and participants who were employed had 7.24 and 2.0 higher odds of being diagnosed with hypertensive crisis, respectively. Married individuals were 3.44 times more likely to be diagnosed with hypertensive crisis compared to those who were single. Participants who were diagnosed with hypertensive crisis had 7.75 higher odds of being hospitalized than those who were not diagnosed.

At multivariable analysis, Individuals with poor compliance to medication had 6.03 higher odds of being diagnosed with hypertensive crisis when compared to those with good compliance. Employed participants were 3.94 times more likely to

**Table 1. Basic demographic and clinical characteristics.**

| Variable | Median, (IQR) OR Frequency (%) | Hypertensive crisis | | P value |
|---|---|---|---|---|
| | | Yes = 185 (18.9%) | No = 792 (81.1%) | |
| **Age In Years** | 36 (26, 49) | 45 (37-59) | 33 (25-45) | **< 0.001** |
| **Age In Years categories** | | | | **< 0.001** |
| <45 | 672 (68.8) | 89 (48.1) | 583 (73.6) | |
| 45-65 | 202 (20.7 | 59 (31.9) | 143 (18.1) | |
| >65 | 103 (10.5) | 37 (20.0) | 66 (8.3) | |
| **Sex** | | | | 0.308 |
| Male | 463 (47.4) | 94 (50.8) | 369 (46.6) | |
| Female | 514 (52.6) | 92 (49.2) | 422 (53.4) | |
| **Employment Status**, n = 968 | | | | **<0.001** |
| Unemployed | 434 (44.8) | 57 (31.49) | 377 (47.9) | |
| Employment | 534 (55.2) | 124 (68.51) | 410 (52.1) | |
| **Chronic Kidney Disease**, n = 972 | | | | 0.138 |
| Yes | 35 (3.6) | 10 (5.4) | 25 (3.2) | |
| No | 942 (96.4) | 175 (94.6) | 767 (96.8) | |
| **Diabetes** | | | | 0.064 |
| Yes | 20 (2.0) | 7 (3.8) | 13 (1.6) | |
| No | 957 (98.0) | 178 (96.2) | 779 (98.4) | |
| **HIV**, n = 881 | | | | 0.755 |
| Positive | 176 (20.0) | 34 (20.9) | 142 (19.8) | |
| Negative | 705 (80.0) | 129 (79.1) | 576 (80.2) | |
| **History of stroke**, n = 968 | | | | **< 0.001** |
| Yes | 14 (1.4) | 11 (6.0) | 3 (0.4) | |
| No | 954 (98.6) | 172 (93.9) | 782 (99.6) | |
| **Hypertensive Medication**, n = 240 | | | | 0.945 |
| Yes | 178 (74.2) | 130 (74.3) | 48 (73.9) | |
| No | 62 (25.8) | 45 (25.7) | 17(26.1) | |
| **Compliance to Medication**, n = 228 | | | | **< 0.001** |
| Good | 85 (37.3) | 43 (25.4) | 42 (71.2) | |
| Poor | 143 (62.7) | 126 (74.6) | 17 (28.8) | |
| **Marital Status**, n = 968 | | | | **< 0.001** |
| Married | 554 (57.2) | 143 (79.0) | 411 (52.2) | |
| Single | 414 (42.8) | 38 (21.0) | 376 (47.8) | |
| **Cigarette Smoking**, n = 976 | | | | 0.734 |
| Yes | 84 (8.6) | 17 (9.2) | 67 (8.5) | |
| No | 892 (91.4) | 167 (90.8) | 725 (91.5) | |
| **Alcohol Consumption**, n = 965 | | | | 0.383 |
| Yes | 301 (31.2) | 62 (33.9) | 239 (29.4) | |
| No | 664 (68.8) | 121 (66.2) | 543 (69.4) | |
| **Hospitalized**, n = 973 | | | | **<0.001** |
| Yes | 444 (45.6) | 151 (82.1) | 293 (37.1) | |
| No | 529 (54.4) | 33 (17.9) | 496 (62.9) | |
| **SBP, Median** mmhg | 126 (112, 143) | | | |
| **DBP, Median** mmhg | 82 (73, 95) | | | |
| **Hypertensive Status** | | | | **<0.001** |
| Yes | 236 (24.2) | 174 (73.7) | 62 (26.3) | |

*(Continued)*

**Table 1.** (Continued)

| Variable | Median, (IQR) OR Frequency (%) | Hypertensive crisis | | P value |
|---|---|---|---|---|
| No | 741 (75.8) | 11 (1.5) | 730 (98.5) | |
| **Type of Hypertensive Crisis** | | | | |
| Emergency | 11 (1.1) | | | |
| Urgency | 174 (17.8) | | | |

**Abbreviation:** IGR: interquartitle range, %: percentage.

**Table 2.** Univariable and Multivariable logistic regression.

| Variable | OR (95%CI) | P-value | AOR (95%CI) | P-value |
|---|---|---|---|---|
| **Age in years** | | | | |
| <45 | ref | | ref | |
| 45-65 | 2.70 (1.85, 3.94) | **<0.001** | 1.59 (0.64, 3.93) | 0.311 |
| >65 | 3.67 (2.31, 5.81) | **<0.001** | 1.73 (0.59, 5.05) | 0.314 |
| **Sex** | | | | |
| Male | ref | | ref | |
| Female | 0.5 (0.61, 1.17) | 0.308 | 1.02 (0.49, 2.10) | 0.964 |
| **Hypertensive status** | | | | |
| No | ref | | | |
| Yes | 186 (96.0, 361.1) | **<0.001** | | |
| **History of Stroke** | | | | |
| No | ref | **<0.001** | ref | |
| Yes | 16.7 (4.60, 60.4) | | 1.93 (0.43, 8.99) | 0.375 |
| **Compliance to medication** | | | | |
| Good | ref | **<0.001** | ref | |
| Poor | 7.24 (3.74, 14.02) | | 6.03 (2.78, 13.01) | **< 0.001** |
| **Employment Status** | | | | |
| Unemployed | ref | | ref | |
| Employment | 2.00 (1.41, 2.82) | **<0.001** | 3.94 (1.52, 10.21) | **0.005** |
| **Marital Status** | | | | |
| Unmarried | ref | | ref | |
| Married | 3.44 (2.34, 5.05) | **<0.001** | 1.16 (0.50, 2.68) | 0.722 |
| **Hospitalized** | | | | |
| No | Ref | | ref | |
| Yes | 7.75 (5.17, 11.56) | **<0.001** | 3.43 (1.61, 7.34) | **< 0.001** |

**Abbreviation:** OR: crude odds ratio, AOR: adjusted odds ratio: Ref: Reference group,

have hypertensive crisis compared to the unemployed. Those who were hospitalized had 3.43 higher odds of being diagnosed with Hypertensive crisis compared to outpatients.

## Hypertensive urgency and hypertensive emergency

We segregated hypertensive crisis by its categories (urgency and emergency) to determine if the correlates of hypertensive crisis were any different, Table 3. Participants diagnosed with hypertensive urgency had a higher median age

**Table 3. Relationship between study variables and hypertensive urgency and hypertensive emergency.**

| Variable | Urgency | | P-value | Emergency | | P-value |
|---|---|---|---|---|---|---|
| | Yes = 17 (17.8%) | No = 803 (82.2%) | | Yes = 11 (1.1%) | No = 966 (98.9%) | |
| **Age In Years** | 45 (36, 59) | 33 (25, 46) | **< 0.001** | 53 (40, 63) | 35 (26, 48) | **0.004** |
| **Age In Years categories** | | | **< 0.001** | | | 0.06 |
| <45 | 85 (48.9) | 587 (73.1) | | 4 (36.4) | 668 (69.2) | |
| 45-65 | 54 (31.0) | 148 (18.4) | | 5 (45.6) | 197 (20.4) | |
| >65 | 35 (20.1) | 68 (8.5) | | 2 (18.2) | 101 (10.5) | |
| **Sex** | | | 0.28 | | | 0.895 |
| Male | 89 (51.2) | 374 (46.6) | | 5 (45.5) | 458 (47.5) | |
| Female | 85 (48.8) | 428 (53.4) | | 6 (54.5) | 507 (52.5) | |
| Employment Status, *n = 968* | | | **< 0.001** | | | 0.570 |
| Unemployed | 53 (31.2) | 381 (47.7) | | 4 (36.4) | 430 (44.9) | |
| Employment | 117 (68.8) | 417 (52.3) | | 7 (63.6) | 527 (55.1) | |
| **Chronic Kidney Disease, *n = 972*** | | | 0.213 | | | 0.323 |
| Yes | 9 (5.2) | 26 (3.2) | | 1 (9.1) | 34 (3.5) | |
| No | 165 (94.8) | 777 (96.8) | | 10 (90.9) | 932 (96.5) | |
| **Diabetes** | | | 0.15 | | | 0.097 |
| Yes | 6 (3.5) | 14 (1.7) | | 1 (9.1) | 19 (2.0) | |
| No | 168 (96.5) | 789 (98.3) | | 10 (90.9) | 947 (98.0) | |
| **HIV, *n = 881*** | | | 0.588 | | | 0.427 |
| Positive | 33 (21.6) | 143 (19.6) | | 1 (10.0) | 175 (20.1) | |
| Negative | 120 (78.4) | 585 (80.4) | | 9 (90.0) | 696 (79.9) | |
| **History of stroke, *n = 968*** | | | **< 0.001** | | | **< 0.001** |
| Yes | 7 (4.1) | 7 (0.9) | | 4 (36.4) | 10 (1.0) | |
| No | 165 (95.9) | 789 (99.1) | | 7 (63.6) | 947 (99.0) | |
| **Hypertensive Medication, *n = 240*** | | | 0.905 | | | 0.667 |
| Yes | 122 (73.9) | 56 (74.7) | | 8 (80.0) | 170 (73.9) | |
| No | 43 (26.1) | 19 (25.3) | | 2 (20.0) | 60 (26.1) | |
| **Compliance to Medication, *n = 228*** | | | **< 0.001** | | | 0.626 |
| Good | 40 (25.2) | 45 (65.2) | | 3 (30.0) | 82 (37.6) | |
| Poor | 119 (74.8) | 24 (34.8) | | 7 (70.0) | 136 (62.4) | |
| **Marital Status, *n = 968*** | | | **< 0.001** | | | 0.666 |
| Married | 136 (80.0) | 418 952.4) | | 7 (65.6) | 547 (57.2) | |
| Single | 34 (20.0) | 380 (47.6) | | 4 (36.4) | 410 (42.8) | |
| **Cigarette Smoking, *n = 976*** | | | 0.74 | | | 0.954 |
| Yes | 16 (9.3) | 68 (8.5) | | 1 (9.1) | 83 (8.6) | |
| No | 157 (90.8) | 735 (91.5) | | 10 (90.9) | 882 (91.4) | |
| **Alcohol Consumption, *n = 965*** | | | 0.182 | | | |
| Yes | 61 (35.5) | 240 (30.3) | | 10 (90.9) | 300 (31.5) | |
| No | 111 (64.5) | 553 (69.7) | | 1 (9.1) | 654 (68.5) | |
| **Hospitalized, n = 973** | | | **< 0.001** | | | **0.015** |
| Yes | 142 (82.1) | 302 (37.8) | | 9 (81.8) | 435 (45.2) | |
| No | 31 (17.9) | 498 (62.3) | | 2 (18.2) | 527 (54.8) | |
| **Hypertensive Status** | | | **< 0.001** | | | **< 0.001** |
| Yes | 164 (94.3) | 72 (9.0) | | 10 (90.9) | 226 (23.4) | |
| No | 10 (5.7) | 731 (91.0) | | 1 (9.1) | 740 (76.6) | |

**Abbreviation:** IGR: interquartile range, %: percentage

compared to those without hypertensive urgency (p<0.001). Participants who were in employment were more likely to experience hypertensive urgency compared to unemployed individuals (p<0.001). Participants with a previous history of stroke were more likely to be diagnosed with hypertensive urgency compared to those without (p<0.001). Among participants diagnosed with hypertensive urgency the majority had Poor compliance to medication (p<0.001). Married individuals were more likely to experience hypertensive urgency compared to single individuals (p<0.001). Hospitalization was more common among those with hypertensive urgency compared to those without hypertensive urgency (p<0.001). Individuals with a history of hypertension were more likely to experience hypertensive urgency compared to those without a history of hypertension (p<0.001).

Participants diagnosed with hypertensive emergency had a higher median age compared to those without hypertensive emergency (p=0.004). Participants with a previous history of stroke were more likely to be diagnosed with hypertensive emergency compared to those without hypertensive emergency (p<0.001). Hospitalization was more common among those with hypertensive emergency compared to those without hypertensive emergency (p=0.015). Individuals with a history of hypertension were more likely to experience hypertensive emergency compared to those without a history hypertension (p<0.001), Table 3.

**Logistic regression analysis of the factors associated with hypertensive urgency and emergency**

For factors associated with urgency at univariable analysis, participants in the age group 45–65 years and above 65 years were 2.52 and 3.55 times more likely to be diagnosed with hypertensive urgency when compared to those aged <45 years, Table 4. Participants with a history of hypertension were 5.57 times more likely to be diagnosed with hypertensive urgency compared to those without a history of hypertension. Individuals with a previous history of stroke had 4.78 increased odds of being diagnosed with hypertensive urgency compared to those without. Participants who had poor compliance to antihypertensive and participants who were employed had 5.58 and 2.02 higher odds of being diagnosed with hypertensive urgency, respectively. Those who were married were 1.00 times more likely to be diagnosed with hypertensive urgency than those who were single. Participants who were diagnosed with a hypertensive urgency were 7.55 times more likely to be hospitalized than those who were not diagnosed with hypertensive urgency.

For factors associated with hypertensive urgency at multivariable analysis, Participants who had poor medication adherence had a 4.92 times greater likelihood of being diagnosed with hypertensive urgency than those with good compliance. Participants who were employed were 2.7 times more likely to experience a hypertensive urgency than those who were unemployed. Those who were hospitalized had 2.08 higher odds of being diagnosed with hypertensive urgency compared to outpatients.

For factors associated with hypertensive emergency at univariable analysis, Individuals aged 45–65 years had significantly higher odds (OR 4.24) of being diagnosed with hypertensive emergency compared to those under 45 years. Participants with hypertension had 32.74 higher odds of being diagnosed with hypertensive emergency compared to those without hypertension. A previous history of stroke increased the odds of being diagnosed with hypertensive emergency by 54.11 compared to those without hypertensive emergency. Participants who were diagnosed with a hypertensive emergency were 5.45 times more likely to be hospitalized than those who were not diagnosed with hypertensive emergency.

On multivariable analysis, no factors were associated with hypertensive emergency.

## Discussion

The study aimed to investigate the prevalence and factors associated with hypertensive crisis. The prevalence of the hypertensive crisis in this study was 18.9%, with 1.1% diagnosed with hypertensive urgency and 17.2% diagnosed with hypertensive emergency. The prevalence of hypertensive crisis in this study (18.9%) was higher than reported in Somalia (2.1%), Tanzania (6.3%), and a systematic review of African studies (9.09%) [3,4,15]. This higher prevalence in our setting is particularly concerning; this finding underscores the need for policymakers in Zambia to invest in public health

**Table 4.** Univariable and Multivariable logistic regression *of the factors associated with hypertensive urgency and emergency.*

| Variable | Hypertensive Urgency | | | | Hypertensive Emergency | | | |
|---|---|---|---|---|---|---|---|---|
| | OR (95%CI) | P-value | AOR (95%CI) | P-value | OR (95%CI) | P-value | AOR (95%CI) | P-value |
| **Age in years** | | | | | | | | |
| <45 | ref | | ref | | ref | | ref | |
| 45-65 | 2.52(1.71, 3.71) | < 0.001 | 0.90 (0.41, 1.93) | 0.731 | 4.24 (1.13, 15.94) | **0.033** | 2.47 (0.55, 11.13) | 0.236 |
| >65 | 3.55 (2.22, 5.67) | < 0.001 | 1.37 (0.50, 3.74) | 0.537 | 3.31 (0.60, 18.29) | 0.17 | 1.45 (0.13, 15.96) | 0.759 |
| **Sex** | | | | | | | | |
| Male | ref | | ref | | ref | | ref | |
| Female | 0.83(0.60, 1.16) | 0.28 | 0.86 (0.45, 1.66) | 0.673 | 1.08 (0.33, 3.58) | 0.895 | 1.03 (0.27, 3.97) | 0.957 |
| **Hypertensive status** | | | | | | | | |
| No | ref | | | | ref | | | |
| Yes | 5.57 (3.03, 10.28) | < 0.001 | – | – | 32.74(4.17, 257.17) | **< 0.001** | – | – |
| **History of Stroke** | | | | | | | | |
| No | ref | | | | ref | | – | – |
| Yes | 4.78 (1.66, 13.82) | **0.004** | – | – | 54.11 (13.65, 214.55) | **< 0.001** | – | – |
| **Compliance to medication** | | | | | | | | |
| Good | ref | | ref | | ref | | ref | |
| Poor | 5.58 (3.03, 10.28) | < 0.001 | 4.92 (2.39, 10.12) | **0.024** | 1.41 (0.35, 5.59) | 0.628 | 1.35(0.29, 6.42) | 0.698 |
| **Employment Status** | | | | | | | | |
| Unemployed | ref | | ref | | ref | | ref | |
| Employment | 2.02(1.42, 2.87) | < 0.001 | 2.7 (1.14, 6.43) | **0.024** | 1.43(0.42, 4.90) | **0.572** | 1.34 (0.19, 9.29) | 0.769 |
| **Marital Status** | | | | | | | | |
| Unmarried | ref | | ref | | ref | | ref | |
| Married | 1.00 (1.13, 6.43) | < 0.001 | 0.92 (0.42, 2.03) | 0.841 | 0.76 (0.22, 2.62) | **0.667** | 2.11 (0.49, 9.18) | 0.317 |
| **Hospitalized** | | | | | | | | |
| No | ref | | ref | | ref | | ref | |
| Yes | 7.55 (4.99, 11.43) | < 0.001 | 2.08 (1.04, 4.15) | **0.038** | 5.45 (1.17, 25.36) | **0.031** | 3.14 (0.36, 26.65) | 0.295 |

**Abbreviation:** OR: crude odds ratio, AOR: adjusted odds ratio: Ref: Reference group.

campaigns to raise awareness about the risks of hypertension and promote healthy lifestyle behaviors, such as regular physical activity, a balanced diet, and smoking cessation, as well as invest in primary healthcare to improve access to hypertension screening and treatment [16,17]. Healthcare professionals should give priority to patient education on preventing hypertension, conducting regular blood pressure checks, initiating antihypertensive medications promptly, and training health personnel on managing hypertensive emergencies [16,18]. In settings like ours with limited resources, these measures can help reduce severe complications such as target organ damage, disabilities, and mortality rates.

In this study, participants diagnosed with hypertensive crisis were predominantly younger, with a nearly equal distribution among males and females. Our study aligns with most studies from Africa which have hypertensive crisis in younger compared to compared to older individuals, while the age and gender distribution of patients with hypertensive crisis differ across various studies [19–21]. Young individuals often perceive themselves as being in excellent health, which can lead to delayed diagnosis and treatment [22]. Limited access to healthcare, often due to a lack of insurance or regular checkups, further exacerbates this issue in our setting [23,24]. Additionally, unhealthy lifestyle choices prevalent among young people such as poor diet, obesity, and substance abuse significantly increase their risk of developing hypertension, a precursor for hypertensive crisis [25].

This present study, aligns with other studies which have found an association between poor compliance among people living with hypertension and development of hypertensive crisis [3,4]. In our setting, poor compliance might probably explained by a lack of knowledge about hypertension and medication adherence, leading to misconceptions among patients [26]. These misconceptions include the belief that hospital medications are harmful, causing them to use unproven local herbs. Secondly, most patients with hypertension and the general population do not seek regular medical checkups and only seek healthcare after developing symptoms suggestive of illnesses [4,26]. Even those few who seek care often present at low level clinics/hospitals without access to specialist healthcare providers and resources to manage hypertension [4,26]. Additionally, the lack of insurance among the majority of patients forces them to depend on government-funded medication, which often has limited supply and few choices tailored to a particular patient's needs [26].

In this current, there was a significant association being employed and being diagnosed with hypertensive crisis. This finding contrasts with study conducted in Ethiopia (2023), which identified a significant relationship between unemployment status and the occurrence of hypertensive crisis [27]. This association may be because certain professions, especially those in sectors like healthcare, social services, administration, education, banking, insurance, transportation, hospitality, and food services, are highly susceptible to work-related stress, which, when prolonged, can contribute to the development of hypertension and associated health complications [28,29]. Hence, there is a clear need to promote strategies for stress management and lifestyle promotion in workplaces to mitigate these risks and improve employee health outcomes [28].

Hypertensive crisis require hospitalization for effective blood pressure control [30,31]. Our study found a strong association between hospitalization and hypertensive crises, emphasizing the importance of an existing policy, which mandates that every person with severely elevated blood pressure is hospitalized to stabilize their blood pressure [32]. However, in our local setting, challenges exist in diagnosing and managing target organ damage among patients who present with hypertensive crisis. These challenges include limited access to essential diagnostic tests to rule out target organ damage for all who presents with elevated blood pressure. This includes laboratory investigations such as liver function tests, kidney function tests, and complete blood counts, as well as radiological investigations like CT scans, Electrocardiogram and ultrasounds. Additionally, a shortage of intravenous antihypertensive medications restricts treatment options. Consequently, complications of hypertensive crisis, such as neurological, cardiovascular, and kidney problems, are likely underdiagnosed in our setting [33,34].

Our study has some strengths and limitations. The main strength is that, to the best of our knowledge and literature search, this is the first study in Zambia to determine the prevalence and associated factors of hypertensive crisis. Thus, it provides valuable insights into the burden of hypertensive crisis and serves as a basis for future research. However, the study also has limitations. It was conducted at a single center, limiting the generalizability of the results, however future research strategies should include multiple centers across different regions to enhance the generalization of the findings. Secondly, due to the retrospective nature of the study, we could not capture all relevant factors associated with a hypertensive crisis, such as lifestyle choices like diet and physical activity, as well as genetic predispositions that may be associated with hypertensive crisis. Thirdly, the study may have been underpowered due to missing information in some variables.

## Conclusion

This study highlights the significant prevalence of hypertensive crisis and identified associated factors in our setting. This burden is associated with high mortality and morbidity coupled with high cost of long-term health care. We therefore need to raise awareness about hypertension through community-based health education campaigns, which include outreach programs like mobile clinics, door-to-door campaigns, and digital outreach through social media, radio, and TV. Efforts to promote medication adherence must include patient-centered approaches, like integrating adherence counseling into routine care and leveraging digital health tools for reminders and follow-ups, which will greatly improve long-term health outcomes.

## Supporting information

**S1 Checklist. Strobe checklist.**
(DOCX)

**S1 Data. Dataset.**
(XLSX)

## Acknowledgments

The authors would like to thank Livingstone University Teaching hospital management for having granted permission to conduct the study and HAND research group for assistance rendered during the study.

## Author contributions

**Conceptualization:** Mbulazi B. Kabeta, Joreen P. Povia, Sepiso K. Masenga.

**Data curation:** Lukundo Siame, Mbulazi B. Kabeta, Joreen P. Povia, Sepiso K. Masenga.

**Formal analysis:** Lweendo Muchaili, Benson M. Hamooya, Sepiso K. Masenga.

**Funding acquisition:** Sepiso K. Masenga.

**Investigation:** Sepiso K. Masenga.

**Methodology:** Lukundo Siame, Ceaser Wankumbu Silumbwe, Sepiso K. Masenga.

**Resources:** Sepiso K. Masenga.

**Supervision:** Sepiso K. Masenga.

**Validation:** Gift C. Chama, Benson M. Hamooya, Bislom C. Mweene, Situmbeko Liweleya, Joreen P. Povia, Sepiso K. Masenga.

**Visualization:** Lukundo Siame, Gift C. Chama, Sepiso K. Masenga.

**Writing – original draft:** Lukundo Siame, Mbulazi B. Kabeta, Gift C. Chama, Lweendo Muchaili, Ceaser Wankumbu Silumbwe, Benson M. Hamooya, Bislom C. Mweene, Situmbeko Liweleya, Sydney Mulamfu, Joreen P. Povia, Sepiso K. Masenga.

**Writing – review & editing:** Lukundo Siame, Mbulazi B. Kabeta, Gift C. Chama, Lweendo Muchaili, Ceaser Wankumbu Silumbwe, Benson M. Hamooya, Bislom C. Mweene, Situmbeko Liweleya, Sydney Mulamfu, Joreen P. Povia, Sepiso K. Masenga.

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
