## [Decision Letter · Decision Letter 0]

28 Oct 2024

PGPH-D-24-01633

Hypertensive Crisis: Insights into Prevalence and associated Factors at a Tertiary Care Facility in Zambia

Dear Dr. Masenga,

Thank you for submitting your manuscript to PLOS Global Public Health. After careful consideration, we feel that it has merit but does not fully meet PLOS Global Public Health’s publication criteria as it currently stands. Therefore, we invite you to submit a revised version of the manuscript that addresses the points raised during the review process.

We look forward to receiving your revised manuscript.

Kind regards,

Yuvaraj Krishnamoorthy

Academic Editor

Journal Requirements:

Additional Editor Comments (if provided):

Reviewers' comments:

Reviewer's Responses to Questions

**Comments to the Author**

1. Does this manuscript meet PLOS Global Public Health’s publication criteria ? Is the manuscript technically sound, and do the data support the conclusions? The manuscript must describe methodologically and ethically rigorous research with conclusions that are appropriately drawn based on the data presented.

Reviewer #1: Yes

Reviewer #2: Yes

2. Has the statistical analysis been performed appropriately and rigorously?

Reviewer #1: Yes

Reviewer #2: Yes

3. Have the authors made all data underlying the findings in their manuscript fully available (please refer to the Data Availability Statement at the start of the manuscript PDF file)?

Reviewer #1: Yes

Reviewer #2: Yes

4. Is the manuscript presented in an intelligible fashion and written in standard English?

Reviewer #1: Yes

Reviewer #2: Yes

5. Review Comments to the Author

Reviewer #1: Significance, Novelty, and Strengths.

The paper addresses a critical health issue, the prevalence of hypertensive crisis, and its associated factors in a significant population at a referral center in Zambia. This is an important contribution to the understanding of hypertension in developing countries, where such data might be scarce.

The study’s originality is demonstrated through its focus on a specific population in Zambia and the use of robust data collection methods from a large sample size (n=977). Additionally, the paper’s approach to using both univariate and multivariable logistic regression analysis to identify factors associated with hypertensive crisis adds depth to the findings.

Methodology: The retrospective cross-sectional design is well-suited to the research question. The systematic sampling method used to select patient records and the clear inclusion and exclusion criteria bolster the study's validity.

Analysis: The use of REDCap for data capture and STATA for statistical analysis ensures robust data handling and analysis. The distinction between univariate and multivariable logistic regression provides comprehensive insights into the factors associated with hypertensive crisis.

Relevance: Given the high prevalence of hypertension in sub-Saharan Africa, this study is highly relevant to public health in this region. The findings have potential implications for clinical practice and policy-making in Zambia and similar contexts.

Weaknesses:

Missing Data: The exclusion of 79 files due to missing data may introduce selection bias. The paper lacks a discussion on how this limitation might affect the study's findings.

Generalizability: The study is conducted in a single referral hospital, which may limit the generalizability of the results to other settings in Zambia or other countries.

Confounding Factors: While the study includes several variables, it does not address potential confounding factors such as diet, physical activity, and genetic predisposition that could influence hypertensive crisis.

Depth of Discussion: The discussion section could be more comprehensive, particularly in contextualizing the findings within the broader literature and discussing the implications for clinical practice and policy.

Suggestions for Improvement:

Address Missing Data: Provide a detailed analysis of the potential impact of the 79 excluded files on the study’s findings. Discuss any patterns in the missing data and use sensitivity analysis to understand how this exclusion might influence the results.

Enhance Generalizability: While acknowledging the limitations of a single-center study, suggest strategies for future research to include multiple centers across different regions to enhance the generalizability of the findings.

Control for Confounders: Incorporate additional variables related to lifestyle and genetic factors in future analyses to control for potential confounding factors. If data on these variables were not available, discuss this limitation and its potential impact on the study's conclusions.

Expand Discussion: Strengthen the discussion section by comparing the study's findings with existing literature from similar contexts. Discuss the implications of the high prevalence of hypertensive crisis for healthcare providers and policymakers in Zambia. Provide more detailed recommendations for practice and policy based on the study’s findings.

Ethical Considerations: Given the retrospective nature of the study, it would be beneficial to discuss how the ethical considerations were managed, particularly concerning patient confidentiality and data security.

Reviewer #2: Areas for Improvement

Confounding Variables: While the multivariable logistic regression approach is appropriate, it is not clear which variables were included as potential confounders in the final model. Providing a rationale for the selection of independent variables or adjusting factors would enhance the transparency of the analysis.

Handling of Missing Data: There is no mention of how missing data, if any, were handled during the analysis. A description of missing data management (e.g., exclusion, imputation, or sensitivity analysis) would strengthen the methodological rigor.

Model Assumptions: It is important to ensure that the assumptions of logistic regression were checked. Consider including diagnostic tests such as multicollinearity assessment (using Variance Inflation Factor) or goodness-of-fit tests (e.g., Hosmer-Lemeshow test).

Clarity in Subgroup Analyses: Some subgroup analyses, such as those related to age and medication adherence, could benefit from clearer statistical justification. If p-values or confidence intervals are available for these comparisons, including them would help highlight the statistical significance of these findings.

3. Methodological Concerns

Definition of Outcome: The primary outcome (hypertensive crisis) is well-defined; however, further clarification on how "target organ damage" was assessed by clinicians would be beneficial for reproducibility. Consider providing examples or criteria for target organ damage if available.

Detailed Description of Study Design: While the study is retrospective and cross-sectional, further details on the data extraction process and criteria for inclusion/exclusion would improve the transparency of the study design.

6. PLOS authors have the option to publish the peer review history of their article (what does this mean? ). If published, this will include your full peer review and any attached files.

**Do you want your identity to be public for this peer review?** For information about this choice, including consent withdrawal, please see our Privacy Policy .

Reviewer #1: **Yes: ** Olivier Mukuku

Reviewer #2: No

---

## [Decision Letter · Decision Letter 1]

26 Nov 2024

PGPH-D-24-01633R1

Hypertensive Crisis: Insights into Prevalence and associated Factors at a Tertiary Care Facility in Zambia

Dear Dr. Masenga,

Thank you for submitting your manuscript to PLOS Global Public Health. After careful consideration, we feel that it has merit but does not fully meet PLOS Global Public Health’s publication criteria as it currently stands. Therefore, we invite you to submit a revised version of the manuscript that addresses the points raised during the review process.

We look forward to receiving your revised manuscript.

Kind regards,

Yuvaraj Krishnamoorthy

Academic Editor

Journal Requirements:

Additional Editor Comments (if provided):

Reviewers' comments:

Reviewer's Responses to Questions

**Comments to the Author**

1. If the authors have adequately addressed your comments raised in a previous round of review and you feel that this manuscript is now acceptable for publication, you may indicate that here to bypass the “Comments to the Author” section, enter your conflict of interest statement in the “Confidential to Editor” section, and submit your "Accept" recommendation.

Reviewer #1: All comments have been addressed

Reviewer #2: All comments have been addressed

2. Does this manuscript meet PLOS Global Public Health’s publication criteria ? Is the manuscript technically sound, and do the data support the conclusions? The manuscript must describe methodologically and ethically rigorous research with conclusions that are appropriately drawn based on the data presented.

Reviewer #1: Yes

Reviewer #2: Yes

3. Has the statistical analysis been performed appropriately and rigorously?

Reviewer #1: Yes

Reviewer #2: Yes

4. Have the authors made all data underlying the findings in their manuscript fully available (please refer to the Data Availability Statement at the start of the manuscript PDF file)?

Reviewer #1: Yes

Reviewer #2: Yes

5. Is the manuscript presented in an intelligible fashion and written in standard English?

Reviewer #1: Yes

Reviewer #2: Yes

6. Review Comments to the Author

Reviewer #1: Lines 31 and 168: In the results section, please revise the presentation of the confidence interval (CI) for the prevalence of hypertensive crisis. Currently, it is reported as:

"The prevalence of hypertensive crisis was 18.9% (95% CI: 0.17, 0.21)."

For better readability and consistency with standard epidemiological reporting formats, we recommend rephrasing it as:

"The prevalence of hypertensive crisis was 18.9% (95% CI: 17.0%, 21.0%)."

Reviewer #2: Ambiguity in Findings:

The high adjusted odds ratio for hypertension (AOR: 29.7; 95% CI: 2.30–384.1) raises concerns about wide confidence intervals, suggesting possible issues with data variability or sample size for this factor.

Distinguish Emergency vs. Urgency:

Analyze and report factors separately for hypertensive emergencies and urgencies, as their management and implications differ significantly.

7. PLOS authors have the option to publish the peer review history of their article (what does this mean? ). If published, this will include your full peer review and any attached files.

**Do you want your identity to be public for this peer review?** For information about this choice, including consent withdrawal, please see our Privacy Policy .

Reviewer #1: **Yes: ** Olivier Mukuku

Reviewer #2: No

---

## [Decision Letter · Decision Letter 2]

3 Jan 2025

PGPH-D-24-01633R2

Hypertensive Crisis: Insights into Prevalence and associated Factors at a Tertiary Care Facility in Zambia

Dear Dr. Masenga,

Thank you for submitting your manuscript to PLOS Global Public Health. After careful consideration, we feel that it has merit but does not fully meet PLOS Global Public Health’s publication criteria as it currently stands. Therefore, we invite you to submit a revised version of the manuscript that addresses the points raised during the review process.

We look forward to receiving your revised manuscript.

Kind regards,

Yuvaraj Krishnamoorthy

Academic Editor

Additional Editor Comments (if provided):

Reviewers' comments:

Reviewer's Responses to Questions

**Comments to the Author**

1. If the authors have adequately addressed your comments raised in a previous round of review and you feel that this manuscript is now acceptable for publication, you may indicate that here to bypass the “Comments to the Author” section, enter your conflict of interest statement in the “Confidential to Editor” section, and submit your "Accept" recommendation.

Reviewer #1: All comments have been addressed

Reviewer #2: All comments have been addressed

2. Does this manuscript meet PLOS Global Public Health’s publication criteria ? Is the manuscript technically sound, and do the data support the conclusions? The manuscript must describe methodologically and ethically rigorous research with conclusions that are appropriately drawn based on the data presented.

Reviewer #1: Yes

Reviewer #2: Yes

3. Has the statistical analysis been performed appropriately and rigorously?

Reviewer #1: Yes

Reviewer #2: Yes

4. Have the authors made all data underlying the findings in their manuscript fully available (please refer to the Data Availability Statement at the start of the manuscript PDF file)?

Reviewer #1: Yes

Reviewer #2: Yes

5. Is the manuscript presented in an intelligible fashion and written in standard English?

Reviewer #1: Yes

Reviewer #2: Yes

6. Review Comments to the Author

Reviewer #1: The authors have made the necessary revisions based on our comments. The article has been significantly improved and now presents clear, coherent, and scientifically robust content. It is ready for publication and deserves to be accepted. No issues related to dual publication, research ethics, or publication ethics have been identified.

Reviewer #2: Methods Section

Insufficient Details:

The description of the retrospective cross-sectional design omits key elements, such as how participants were selected and whether records with missing data were excluded.

The methodology for identifying target organ damage is unclear. Was it based solely on the attending clinician’s judgment?

Potential Bias: The abstract does not address potential biases inherent in a retrospective design, such as data quality issues or missing information.

Statistical Analysis:

The explanation of multivariable logistic regression is too brief. There’s no indication of how variables were selected for inclusion or how confounding was managed.

Handling of outliers (e.g., extremely high blood pressure values) is not mentioned.

Results Section

Incompleteness:

While the prevalence is reported, the clinical or public health implications of hypertensive crisis prevalence are not explored.

The age and gender distribution are described but lack analysis. Why were younger individuals predominantly affected?

Overemphasis on Statistical Significance: The mention of p-values is necessary, but the abstract does not interpret the practical significance of the findings (e.g., how employment or hospitalization contributes to hypertensive crisis).

Superficial Reporting: The distinction between hypertensive urgency (17.8%) and emergency (1.1%) is reported without elaboration on their clinical significance.

Conclusion Section

Repetition: The conclusion repeats the results (e.g., prevalence and associated factors) instead of summarizing the broader implications for practice or policy.

Generic Recommendations: While public awareness and medication adherence are emphasized, no specific strategies or next steps are provided, limiting the conclusion's utility.

Overgeneralization: The abstract claims hypertensive crisis is a "significant proportion" without specifying what this means in a broader context, such as the burden on healthcare resources or patient outcomes.

General Issues

Wordiness: The abstract could be more concise by removing repetitive phrases and focusing on the most critical findings.

Clarity: Key terms such as "employment" and "hospitalization" are mentioned as risk factors but lack context. Why these variables are significant is not addressed.

Engagement: The abstract does not effectively highlight the urgency or relevance of the findings to the healthcare system or public health.

Suggestions for Improvement

Clearly articulate the study's significance in the background without referencing external comparisons.

Provide more methodological transparency, including criteria for target organ damage and handling of missing or incomplete records.

Focus on the key findings without excessive repetition, ensuring that each section adds value.

Strengthen the conclusion with actionable recommendations rather than merely reiterating results.

7. PLOS authors have the option to publish the peer review history of their article (what does this mean? ). If published, this will include your full peer review and any attached files.

**Do you want your identity to be public for this peer review?** For information about this choice, including consent withdrawal, please see our Privacy Policy .

Reviewer #1: **Yes: ** Olivier Mukuku

Reviewer #2: No

---

## [Decision Letter · Decision Letter 3]

27 Mar 2025

PGPH-D-24-01633R3

Hypertensive Crisis: Insights into Prevalence and associated Factors at a Tertiary Care Facility in Zambia

Dear Dr. Masenga,

Thank you for submitting your manuscript to PLOS Global Public Health. After careful consideration, we feel that it has merit but does not fully meet PLOS Global Public Health’s publication criteria as it currently stands. Therefore, we invite you to submit a revised version of the manuscript that addresses the points raised during the review process.

The revision was evaluated by two previous reviewers. One of the reviewer still raised some questions. Please revise the manuscript carefully.

We look forward to receiving your revised manuscript.

Kind regards,

Jianhong Zhou

Staff Editor

Journal Requirements:

Additional Editor Comments (if provided):

Reviewers' comments:

Reviewer's Responses to Questions

**Comments to the Author**

1. If the authors have adequately addressed your comments raised in a previous round of review and you feel that this manuscript is now acceptable for publication, you may indicate that here to bypass the “Comments to the Author” section, enter your conflict of interest statement in the “Confidential to Editor” section, and submit your "Accept" recommendation.

Reviewer #1: All comments have been addressed

Reviewer #2: All comments have been addressed

2. Does this manuscript meet PLOS Global Public Health’s publication criteria ? Is the manuscript technically sound, and do the data support the conclusions? The manuscript must describe methodologically and ethically rigorous research with conclusions that are appropriately drawn based on the data presented.

Reviewer #1: Yes

Reviewer #2: Yes

3. Has the statistical analysis been performed appropriately and rigorously?

Reviewer #1: Yes

Reviewer #2: Yes

4. Have the authors made all data underlying the findings in their manuscript fully available (please refer to the Data Availability Statement at the start of the manuscript PDF file)?

Reviewer #1: Yes

Reviewer #2: Yes

5. Is the manuscript presented in an intelligible fashion and written in standard English?

Reviewer #1: Yes

Reviewer #2: Yes

6. Review Comments to the Author

Reviewer #1: (No Response)

Reviewer #2: Abstract

result

Lack of

"adherence to hypertension medication (adjusted odds ratio (AOR): 5.1; 95% CI: 2.41-

9.78; p < 0.001), having hypertension (AOR: 29.7; 95% Cl 2.30-384.1, p=0.009), being

in employment (AOR: 3.80; 95% CI 1.53-9.46) and hospitalization (AOR: 3.48; 95% CI:

1.61-7.49; p<0.001) were positively associated with hypertensive crisis"

this kind of reporting can confused readers. example those with employment were 3.8 times more like to adhere to medication compare to.....................: this will give clear pictures of what you want to report

conclusion

thee is no need to report , "with a prevalence of 18.9% (189

per 1,000 patients)"this in your abstract conclusion.

methods and materials

Sample size

91 The estimated minimal sample size was 1007. We estimated a prevalence of 6.3 % from

92 a study conducted in Tanzania [4]. The alpha level of 1.5% and design effect of 1. We

93 used openepi.com software to calculate the sample size.

show us the formulary that you used in this work. how did you apply the mathematical formula

result

Relationship between hypertensive crisis and other covariates. the authors used p-value so the authors should should re-interpret the result again and they should stop using the percentage. example Age (<0.001), employment (0.002) were significantly associated with.....................

7. PLOS authors have the option to publish the peer review history of their article (what does this mean? ). If published, this will include your full peer review and any attached files.

**Do you want your identity to be public for this peer review?** For information about this choice, including consent withdrawal, please see our Privacy Policy .

Reviewer #1: **Yes: ** Olivier Mukuku

Reviewer #2: No

---

## [Editor Report · Decision Letter 4]

28 Apr 2025

Hypertensive Crisis: Insights into Prevalence and associated Factors at a Tertiary Care Facility in Zambia

PGPH-D-24-01633R4

Dear Prof. Masenga,

We are pleased to inform you that your manuscript 'Hypertensive Crisis: Insights into Prevalence and associated Factors at a Tertiary Care Facility in Zambia' has been provisionally accepted for publication in PLOS Global Public Health.

Best regards,

Julia Robinson

Executive Editor